# Long-Term Results in Minimally Invasive Non-Resectional Mitral Valve Repair for Barlow Mitral Valve Disease

**DOI:** 10.3390/jcm14031005

**Published:** 2025-02-05

**Authors:** Nicola A. Koch, Jonas Chiappini, Lisa M. Ihringer, Andrei A. M. Caraconi, Islam Salikhanov, Brigitta Gahl, Denis Berdajs

**Affiliations:** 1Department of Cardiac Surgery, University Hospital Basel, CH-4031 Basel, Switzerlandjonas.chiappini@stud.unibas.ch (J.C.); lisa-marie.ihringer@student.uni-tuebingen.de (L.M.I.); andrei.caracioni@stud.medunigraz.at (A.A.M.C.); islam.salikhanov@usb.ch (I.S.); brigitta.gahl@usb.ch (B.G.); 2Surgical Outcome Research Center Basel, University Hospital Basel, University Basel, CH-4031 Basel, Switzerland

**Keywords:** mitral valve, minimally invasive valve surgery, non-resectional mitral valve repair, chordae replacement

## Abstract

**Objective**: The aim was to assess the long-term outcomes, safety, and durability of total endoscopic mitral valve repair for Barlow mitral valve disease. **Methods**: A retrospective analysis of 98 patients undergoing minimal invasive total endoscopic mitral valve repair for Barlow mitral valve disease was conducted between May 2009 and December 2023. A non-resectional repair approach using artificial neochordae and/or ring annuloplasty was performed. Clinical and echocardiographic follow-ups were completed. Rates per patient-years with 95% confidence intervals (CI) for all time-to-event outcomes were calculated. **Results**: The mean age was 59 ± 12, and 43% were female. Minimally invasive mitral valve repair was successfully performed in all 98 patients with no conversions to sternotomy or mitral valve replacement. There was no mitral valve-related reoperation during the hospital stay. Procedural safety was as follows: no in-hospital mortality, no stroke, and no perioperative myocardial infarction. The mean follow-up was 4.1 ± 3.1 years. Survival at seven years was 87% (95% CI 63% to 96%). Freedom from myocardial infarction, stroke, and congestive heart failure was 89% (95% CI 60% to 97%), 93% (95% CI 82% to 97%), and 100%, respectively. Recurrent mitral valve insufficiency at Grade ≥ 2 was diagnosed in *n* = 4 (4.1%) of cases. **Conclusions**: Minimally invasive mitral valve repair using a non-resectional technique for Barlow disease can be performed with a low complication rate. The total endoscopic approach is safe in the long term, with minimal risk of reoperation and recurrent mitral valve insufficiency.

## 1. Introduction

In developed countries, mitral valve regurgitation (MR) is one of the most prevalent valvular heart disorders. The primary causes encompass degenerative pathology (60–70%), followed by ischemic mitral valve regurgitation (20%), endocarditis (2–5%), and rheumatic disease (2–5%) [1]. Surgical mitral valve repair, compared to valve replacement, significantly enhances outcomes and diminishes mortality rates in patients with severe MR by approximately 70%. In asymptomatic patients, optimal surgical outcomes at mid- and long-term can be achieved in specialized centers with low operative mortality (<1%) and high repair rates (≥80–90%).

Mitral valve repair poses challenges, especially in Barlow’s disease, characterized by an excess of leaflet tissue and annular dilatation [2,3]. The surgical approaches vary from the well-accepted resectional technique proposed by Carpentier [4,5,6] to the edge-to-edge approach introduced by Alfieri [7], and the non-resectional approach involving neochordae implantation [8,9]. Often, a combination of these techniques is employed to ensure optimal valve function [10].

The non-resectional approach, employing ring annuloplasty with or without neochordae implantation in Barlow’s disease, has shown promising results. In cases of balanced prolapse with a central jet, isolated ring annuloplasty could also yield satisfactory long-term results [11,12]. However, long-term data on the non-resectional approach for severe Barlow mitral valve insufficiency, along with information on repair durability, remain scarce.

Our institution has amassed over two decades of experience in minimally invasive mitral valve repair utilizing a non-resectional approach comprising neochordae implantation and/or ring annuloplasty. We herein present our long-term results on non-resectional mitral valve repair in a contemporary cohort of patients with Barlow’s disease and severe mitral valve regurgitation, focusing on procedural outcomes, success rates, and safety profiles.

## 2. Materials and Methods

### 2.1. Ethics Statement

The study protocol was approved by the local ethical committee at the University of Basel, Switzerland (EKNZ 2021-00454). Given the retrospective nature of the study, the requirement for written informed consent specific to this study was waived.

### 2.2. Patient Population

A total of 98 consecutive patients with Barlow mitral valve pathology who underwent non-resectional minimally invasive total endoscopic mitral valve repair between September 2009 and December 2023 were included. All patients exhibited evidence of severe or moderate mitral valve regurgitation, primarily attributed to isolated bileaflet prolapse or combined flail leaflets at the posterior and/or anterior leaflets due to chordal elongation and/or rupture.

Exclusion criteria for the study included patients with rheumatic valve disease, patients with active valve endocarditis, or those undergoing concomitant procedures. Additionally, patients with calcification of abdominal aorta and/or the iliac vessels or presence of vulnerable plaques exceeding 3 mm in the thoracic and abdominal aorta diagnosed via computed tomography angiogram (CT) were deemed ineligible for the minimally invasive approach [13].

Patients’ data were obtained from our institutional database (Intellect 1.7, Dendrite Clinical Systems, Henley-on-Thames, UK), which is checked for completeness and consistency monthly. Follow-up assessments were conducted based on information obtained from the outpatient clinic register or through telephone calls to patients and the referring physician’s office. Postoperatively, transthoracic echocardiography (TTE) was performed at discharge and at a follow-up of six months after the surgical intervention. Further TTE assessments were carried out at the discretion of the referring cardiologist.

### 2.3. Surgical Technique

The technical aspects of minimally invasive total endoscopic mitral valve repair at our institution have been described previously [14]. Briefly, patients were positioned in a 30° right supine position, and cardiopulmonary bypass was established using femoral vessel cannulation. A 4 to 5 cm periareolar skin incision was made beneath the nipple for both male and female patients. Access to the fourth intercostal space at the mid-clavicular line was obtained using a soft tissue retractor (Alexis Wound-Protector, Applied Medical, Santa Margarita, CA, USA). The scope access was through the fourth intercostal space at the anterior axillary line (Aesculap, Tuttlingen, Germany).

Once cardiopulmonary bypass was initiated and cardioplegia was successful, an atrial retractor (Geister, Tuttlingen, Germany) was introduced through the fourth intercostal space medial to the thoracic incision.

Mitral valve pathology was diagnosed based on echocardiographic findings and intraoperative valve analysis according to the following features: single or bileaflet prolapse with elongated and/or ruptured chordae tendineae, functional prolapse with typical leaflet billowing, and/or presence of excessive tissue or indentations. The surgical approach, ranging from simple annuloplasty to annuloplasty with application of artificial chordae, was decided after surgical analysis of the valve. In case of bileaflet billowing and a central regurgitation jet, ring-only repair was performed. In patients with bileaflet billowing and an eccentric regurgitation jet due to chordae prolapse, ring annuloplasty and chordal replacement were adapted. For annuloplasty, complete semi-rigid rings were used. The size of the mitral valve ring was determined based on the length of the anterior leaflet and the intertrigonal distance. For chordal replacement, PTFE (polytetrafluoroethylene) Seramon^®^ (Serag-Wiessner, Naila, Germany) loops were used. Free-hand artificial neo-chordal replacement was performed using 4/0 ePTFE sutures (Gore-TEX CV4, Flagstaff, AZ, USA) [14].

For atrial fibrillation, left atrial ablation and closure of the left atrial appendage were performed. Patients who were elderly, had longstanding persistent atrial fibrillation, or enlarged left atria were not considered suitable candidates for rhythm correction therapy.

Clinical Parameters: In-hospital mortality was defined as death occurring before discharge. A neurologic event was defined according to the criteria set by the Valve Academic Research Consortium [15]. Perioperative stroke was defined as any neurological deficit with or without evidence of cerebral injury on a CT scan and/or MRI. Perioperative myocardial infarction was defined in accordance with the Universal Definition of Myocardial Infarction.

Definition of Combined Adverse Cardiovascular Events: Combined major adverse cardiovascular events (MACE) during the hospital stay were defined as a combination of in-hospital mortality, stroke, and myocardial infarction. MACE during follow-up was defined as a combined event comprising stroke, myocardial infarction, mortality, and the incidence of congestive heart failure.

Primary outcomes were as follows: (a) all-cause mortality during the observation period, (b) freedom from recurrence of Grade ≥2 mitral valve regurgitation, and (c) freedom from mitral valve-related reoperation during follow-up.

Operative success was defined as a successful primary mitral repair without the need for conversion to valve replacement and without any mitral valve-related reoperation within the first 30 days.

Perioperative safety was defined as the absence of death, perioperative myocardial infarction, stroke, or reoperation for bleeding within the perioperative period (30 days).

Secondary outcomes: Included the incidence of MACE during the hospital stay and throughout the follow-up.

Statistical methods: For assessing patient outcomes during follow-up in terms of event rates, rates per patient-years with 95% confidence intervals (CI) were calculated. Kaplan-Meier methods were employed for visualization, with right-censoring at ten years post-surgery. Pointwise confidence bands for the survival estimate of the study cohort were calculated. Mixed linear models, incorporating time since surgery and preoperative left ventricular ejection fraction (LVEF) as fixed effects and patient number as a random effect, were used to analyze iterative assessments of LVEF during the follow-up period. Multilevel ordered logistic regression, with time and the grade of mitral valve insufficiency at baseline as fixed factors and patient number as random factor, was used for assessing mitral valve insufficiency. Continuous data were presented as the mean ± standard deviation for normally distributed data and as the median with interquartile ranges for skewed distributions. Categorical variables were presented as numbers with percentages. All statistical analyses were performed using Stata 16 (Stata Corp, College Station, TX, USA).

## 3. Results

### 3.1. Early Outcomes

Out of 478 minimally invasive mitral valve repairs, 98 patients underwent mitral valve repair for Barlow disease. The mean age was 59 ± 12 years, and 43% *(n* = 42) were female. Patients’ demographics are presented in Table 1. The mean body mass index (BMI) was 24 ± 3.6 kg/m^2^, 4.1% (*n* = 4) cases have a history of stroke, 3.1% (*n* = 3) had COPD, 2% (*n* = 2) had a history of myocardial infarction, 44%(*n* = 45) had hypertension, and 19% (19) had atrial fibrillation. The median EuroSCORE 2 was 0.87% (IQR 0.64 to 1.3), and 26% (*n* = 25) of patients were classified as New York Heart Association class III/IV heart failure.

The left ventricular ejection fraction was 60 ± 8.8%, and Barlow disease was confirmed in all 98 patients. A history of endocarditis was diagnosed in 2% (*n* = 2) of cases, and both patients underwent surgery following the completion of conservative therapy. In both post-endocarditis cases, mitral valve insufficiency was moderate preoperatively, while severe mitral valve insufficiency was the primary reason for surgical intervention in 98.9% (*n* = 97) of cases (Appendix A).

On average, 1.7 ± 0.83 neochords were implanted per mitral valve repair, with a mean ring size of 38 ± 2.4 mm. In 41.8% (*n* = 41) of cases, mitral valve insufficiency resulted from balanced bileaflet prolapse with a central jet, leading to isolated ring annuloplasty, with a mean ring size of 38 ± 1.9 mm. Ring annuloplasty combined with neochordae implantation was performed in 58.1% (*n* = 57) of cases. In 44.8% (*n* = 44) of cases, neochords were attached to the posterior leaflet, in 6.1% (*n* = 6) to the anterior leaflet, and in 7.1% (*n* = 7) to both leaflets (Appendix A).

Concomitant procedures included closure of the left atrial appendage with left atrial ablation in 4.1% (*n* = 4) cases, closure of the anterolateral commissure in 2.0% (*n* = 2) of cases, and closure of the posterolateral commissure in 2.0% (*n* = 2) of cases. Surgical success was satisfactory, with no instances of conversion to mitral valve replacement or mitral valve repair-related reoperation within 30 days post-discharge. There were no incidents of systolic anterior motion (SAM). The perioperative results are presented in Table 2. Perioperative safety was 91.8%, with no in-hospital mortality, no incidence of stroke, and eight patients (8.1%) requiring reoperation due to bleeding. The incidence of major adverse cardiac events (MACE) during the hospital stay was 0% (*n* = 0), while 23% (*n* = 23) of patients developed atrial fibrillation at discharge. The median stay in the intensive care unit (ICU) was 1.5 days (IQR: 1 to 2 days), and the median hospital stay was 8.0 days (IQR 7.0 to 10 days). Permanent pacemakers were implanted in 4.2% (*n* = 3) of cases.

### 3.2. Late Outcomes

The mean follow-up time was 4.1 ± 3.1 years and was completed in all patients. Each patient underwent at least one postoperative echocardiographic assessment of the mitral valve. During the follow-up period, a total of seven major adverse cardiac events (MACE) were recorded. Specifically, there were three cases of mortality, four strokes, and four myocardial infarctions, with no instances of patients being hospitalized due to congestive heart failure (Table 3).

Freedom from MACE at 3 and 7 years was 95% (95% CI 87% to 98%) and 91% (95% CI 80% to 96%), respectively (Figure 1). Survival probability at 3 and 7 years was 100% and 94% (95% CI 75% to 99%), respectively (Figure 2). Freedom from myocardial infarction, stroke, and congestive heart failure at 3 and 7 years was as follows: (a) 97% (95% CI 90% to 99%) and 97% (95% CI 90% to 99%) for myocardial infarction, (b) 97% (95% CI 89% to 99%) and 93% (95% CI 84% to 98%) for stroke, and (c) 100% for congestive heart failure. (Appendix A).

All cases underwent echocardiographic follow-up assessments, revealing that the mean left ventricular ejection fraction post-surgery was 55.7 ± 8.9% and remained stable throughout the follow-up period. Of all patients in the cohort, 81% had no or mild mitral valve regurgitation during the entire follow-up period, 13% had moderate regurgitation, and 6% had severe regurgitation.

Changes in mitral valve insufficiency were noted during follow-up, with moderate mitral valve regurgitation registered in 18 cases. Four cases required mitral valve reoperation by the end of the follow-up period, primarily due to recurrent severe mitral valve regurgitation. The risk of worsened repaired valve insufficiency increased slightly over time, with an odds ratio of 1.2 per year of follow-up. None of the cases involved endocarditis. During follow-up, LVEF, in comparison to the preoperative value, increased insignificantly by 0.24 (95%CI −0.15 to 0.62) per year. However, the probability of mitral insufficiency increased by 1.22 (95% CI 1.08 to 1.38) per year during follow-up.

Freedom from mitral valve-related reoperation at 3 and 7 years was 97% (95% CI 91% to 99%) and 95% (95% CI 85% to 98%), respectively (Figure 3).

## 4. Discussion

In our series, we demonstrate that the total endoscopic minimally invasive approach for mitral valve repair in Barlow disease is safe, with very good in-hospital and long-term results. In complex Barlow pathologies, a non-resectional approach was successfully applied. In the majority of cases, severe mitral valve regurgitation with pronounced segmental leaflet prolapse due to chordal rupture or prolongation was treated with annuloplasty and artificial loop chordae implantation. In almost 41.8% of cases (*n* = 41), a balanced type of Barlow disease with a multiple prolapse pattern and central jet was present. In those cases, isolated ring annuloplasty for valve regurgitation was performed. No further manipulation of leaflets was conducted, such as, for example, an edge-to-edge stitch or closure of the false commissure. Our success was defined as the combined endpoint of freedom from conversion to valve replacement, conversion to sternotomy, as well as the absence of any mitral valve-related reoperation during the first 30 days. This combined endpoint was achieved in all cases.

The results should be analyzed in more detail. Our philosophy of non-resectional mitral valve repair in Barlow disease is based on two concepts described in the past. Namely, in cases of well-isolated prolapse due to chordal rupture and/or elongation, the “respect rather than resect” approach was adopted [15], which includes the application of pre-measured neochord loops and ring annuloplasty [16]. On the other hand, in a balanced pattern of Barlow mitral valve disease with a central regurgitation jet and absence of leaflet prolapse, an isolated ring annuloplasty was adopted as the repair method.

Our decision-making process for choosing one approach over another is fundamentally based on echocardiographic findings and intraoperative analysis of valve pathology. In isolated balanced pathology with a central regurgitation jet, a ring annuloplasty was the therapy of choice, whereas in cases with an eccentric jet and evidence of leaflet prolapse due to the chordal pathology, annuloplasty and the chordal loop technique was applied. This was done with goal of minimizing tension on any part of the mitral valve apparatus as much as possible and achieving adequate coaptation line in the middle of the mitral valve orifice.

In unbalanced pathology, the leaflet prolapse is primarily addressed by restoring the anatomy of the primary chordae. The length of adjacent primary chordae originating from the same papillary muscle served as a reference. Slight undersizing of the implanted loop was performed with the objective of positioning the free edge of the flail segment below the non-prolapsing leaflet, rather than in the original description of the non-resectional approach where the free edge of the prolapsed segment was displaced into the ventricle [15].

Our philosophy aimed to treat the flail segment without limiting its motion. Instead of pulling the free edge of the prolapsing leaflet completely into the left ventricle, we positioned it at the midpoint between the papillary muscle and the annular plane of the valve. This approach preserved the mobility of the corrected segment and maintained coaptation without displacing it from the midline of the mitral valve ostium.

A similar concept of coaptation restoration was applied in cases of balanced bileaflet prolapse and a central jet. In Barlow disease, the presence of a large annulus with pathological movement in late systole is a significant factor contributing to the billowing of the mitral valve above the annular level, leading to valve regurgitation [9,11,12]. Additionally, the decoupling of annular and ventricular contraction, along with paradoxical movement of the subvalvular apparatus towards the annulus during systole, further exacerbates mitral valve incompetence in systole [9,17,18].

It appears that annular stabilization may address these pathological phenomena, and isolated ring annuloplasty may lead to the remodeling of mitral valve function and could be considered a sufficient repair approach [11,12,19]. An annuloplasty-only procedure stabilizes annular movement and repositions the billowing leaflets beneath the annular level; consequently, the coaptation zone is also relocated underneath the annulus. Our results suggest that the isolated ring annuloplasty provides a valid long-term solution for balanced bileaflet prolapse. Nevertheless, it necessitates a meticulous decision-making process concerning ring size selection to mitigate the risk of SAM, which in the literature may be as high as 5 to 10% [11,12,19]. In our series, the ring size, which was in average 38 ± 2.4 mm, was based on matching it to the anterior leaflet plus one size. This slight “oversizing” strategy promoted balanced coaptation at the midpoint of the mitral valve ostium, ensuring symmetrical force distribution at the coaptation zone. Consequently, the risk of SAM due to the push effect of the posterior leaflet was effectively minimized [19,20], and no cases of SAM were observed in our cohort. We believe that the use of a larger ring without tissue resection may offer advantages in cases of failed mitral valve ring-only repair when considering MitraClip as a solution. In this context, the non-resected valve tissue can serve as a solid anchoring substrate. Additionally, the utilization of a relatively large ring size can help prevent valve stenosis following MitraClip implantation. These advantages may be less pronounced in cases where valve resection and annulus plication are performed, leading to the application of smaller rings.

Our results may also be compared to the recent literature [21,22], where different concepts of mitral valve repair in Barlow disease are analyzed [21,22]. Polzl et al. provides 5-year results of minimally invasive repair in a highly selected low-risk population, composed of male patients. Similar to our results, their findings provide excellent surgical outcomes. Indeed, our reoperation rate for bleeding (8%) is relatively high in compariso, and may be in part explained by avery conservative anticoagulation management. However, our surgical mortality, conversion to the mitral valve replacement, stroke rate, and incidence of major cardiovascular events stay in line with published data from larger cohorts [19,21,22,23].

This is a single-center, retrospective study focusing on surgical outcomes in complex mitral valve pathology. Completeness of clinical and regular echocardiographic follow-up provides reliable long-term results and reflects outcomes in this specific cohort. The results presented strongly suggest that minimally invasive non-resectional mitral valve repair for Barlow disease is safe and offers good long-term durability. The minimally invasive non-resectional approach is portrayed as a simple and effective solution for addressing this complex pathology.

## 5. Limitations

The lack of a control group that would compare the resectional versus non-resectional approach for similar mitral valve pathologies is a notable limitation. The single-center, retrospective approach with a highly selected patient cohort is one of the disadvantages. Left ventricular function was preserved, but we do not have information on left ventricular remodeling during follow-up. Additionally, the lack of information on left ventricular remodeling following reconstruction, as well as long-term medication use, should be considered when interpreting the results.

## Figures and Tables

**Figure 1 jcm-14-01005-f001:**
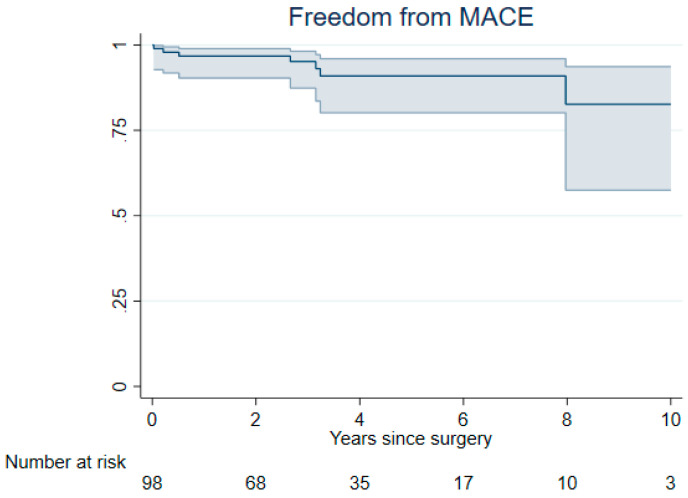
Kaplan-Meier plot for freedom from MACE in patients following minimally invasive non-resectional mitral valve repair in Barlow disease, and was 95% (95% CI 87% to 98%) at 3 years and 91% (95% CI 80% to 96%) at 7 years.

**Figure 2 jcm-14-01005-f002:**
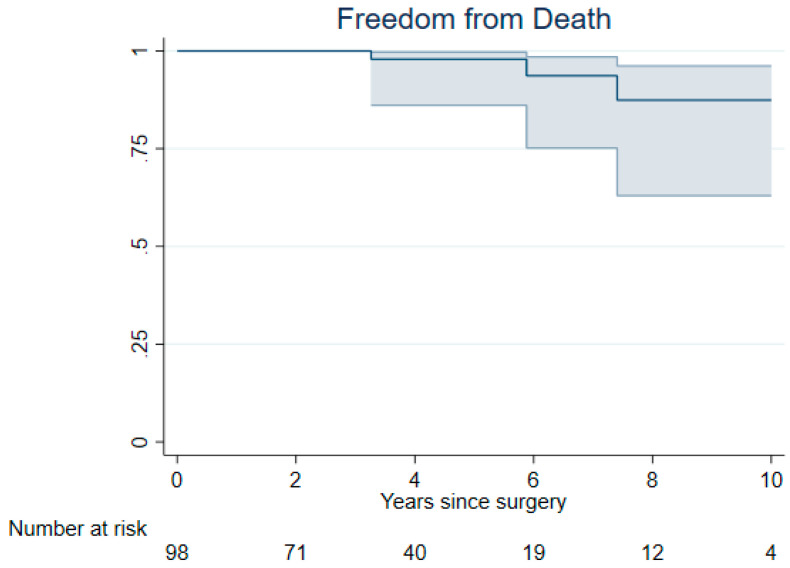
Kaplan-Meier plot for freedom of all cases of mortality for patients undergoing minimally invasive non-resectional mitral valve repair in Barlow disease was 100% at 3 years and 94% (95% CI 75% to 99%) at 7 years.

**Figure 3 jcm-14-01005-f003:**
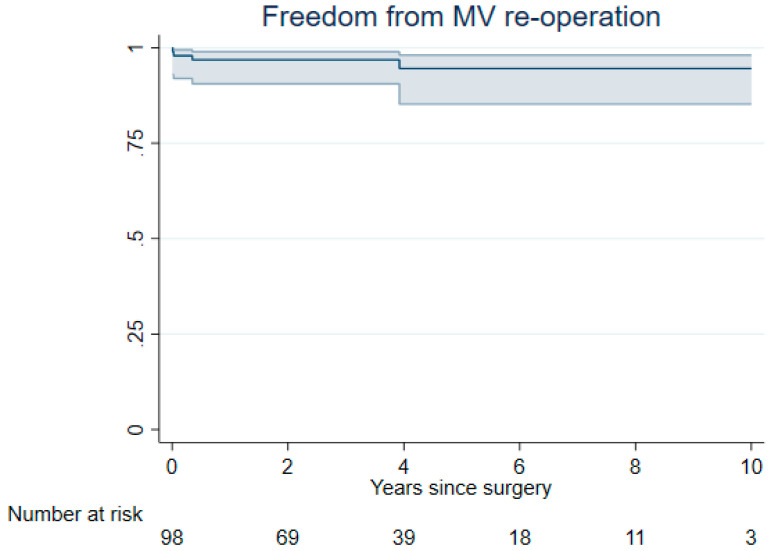
Kaplan-Meier plot for freedom from mitral valve-related reoperation during follow-up in patients following minimally invasive non-resectional mitral valve repair in Barlow disease, which was 97% (95% CI 91% to 99%) at 3 years and 95% (95% CI 85% to 98%) at 7 years.

**Table 1 jcm-14-01005-t001:** Baseline clinical characteristics.

	Total (*N* = 98)
Age	59 (12)
Gender	
Female	42 (43%)
Male	56 (57%)
BMI	24 (3.6)
Current smoker	3 (3.1%)
Preoperative stroke	4 (4.1%)
History of myocardial infarction	2 (2%)
Renal impairment	2(2%)
COPD	2 (2.8%)
Hypertension	44 (45%)
NYHA III or IV	25 (26%)
Atrial fibrillation	19 (19%)
EuroSCORE 2	0.87 (0.64 to 1.3)

NYHA: New York Heart Association; COPD: chronic obstructive pulmonary disease: BMI: body mass index. Values are *n* (%) for categorical variables or median (interquartile range) for continuous variables.

**Table 2 jcm-14-01005-t002:** Postoperative outcomes.

	Total (*N* = 98)
In-hospital mortality	0 (0%)
Cardiac-related death	0 (0%)
Reoperation for bleeding	8 (8.2%)
Atrial fibrillation at discharge	23 (23%)
Pulmonary infection	3 (3.1%)
Postoperative MI	0 (0%)
Postoperative stroke	0 (0%)
Postoperative MACE	0 (0%)
Postoperative renal failure	1 (1.0%)
Renal substitution therapy	0 (0%)
Intubation > 72 h	0 (0%)
Length of ICU stay	2.0 (1.0 to 3.0)
Length of hospital stay	8.0 (7.0 to 10)
Need for pacemaker implantation	5 (5.1%)

Values are n (%) for categorical variables or median (interquartile range) for continuous variables.

**Table 3 jcm-14-01005-t003:** Incidence of individual major adverse cardiovascular events during follow-up.

MACE	Number of Events	Person Years	Rate per Patient-Year (95% CI)
Death	3	399	0.75 (0.24 to 2.3)
Myocardial infarction	4	382	1.05 (0.39 to 2.8)
Stroke	4	384	1.04 (0.39 to 2.8)
Congestive heart failure	0	399	0.00 (0.00 to 0.8)
Any MACE	7	371	1.89 (0.90 to 4.0)

CI: confidence interval; MACE: major adverse cardiovascular event.

## Data Availability

Raw data were generated at the University Hospital of Basel Department of Cardiac Surgery. Derived data supporting the findings of this study are available from the corresponding author on request.

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
