# Peer review of "Long-Term Results in Minimally Invasive Non-Resectional Mitral Valve Repair for Barlow Mitral Valve Disease"

_jcm, 2025, doi:10.3390/jcm14031005_

Round 1

Reviewer 1 Report

Comments and Suggestions for Authors

The manuscript is very interesting and well written.

Authors made a retrospective study on minimally invasive non resectional mitral valve repair for Barlow mitral valve disease.

Review Point-by-point:

  • the manuscript is clear, relevant in his field and presented in a well-structured manner.  
  • The cited references are mostly recent publications and all relevant for the paper.
  • The manuscript scientifically sound and the design are appropriate to test the hypothesis.
  • The manuscript’s results are reproducible based on the details given in the methods section.
  • Figures and tables are appropriate and properly show the data.
  • The conclusions are consistent with the evidence even if discussion should be focused more on results than on authors opinion and surgical choose.
  • Ethics statements and data availability are adequate.

Results

-        Line 166: 34 female and 47% doesn’t fix with the total number of patients (98).

-        Line 181-184: 50 cases of neochordae implantation but with 40 cases of posterior, 4 of anterior and 16 of both leaflets. (40+4+16= 60).

-        Table 1: n° 42 (43%) female + 38 (53%) male à total of 80

Discussion

-        It seems that the discussion section is more an explanation of yours surgical strategy rather than an analysis of your results.

Limitations

-        Line 325-332: these are not limitations of the study, but they are instead the strong points; so they should be removed from this section.

-        I would had some limitations:

o   Patients are hyper selected (mean age 59, mean Euroscore 0,87%, low comorbidity)

o   Follow-up of last patients don’t reach the 7 years.

Author Response

Comment 1 Line 166: 34 female and 47% doesn’t fix with the total number of patients (98). 

Answer: Thank you very much for your valuable comment, it was a typo indeed 43% were females what corresponds to n=42 patients. This was corrected now on P6 L148

Comment 2: Line 181-184: 50 cases of neochordae implantation but with 40 cases of posterior, 4 of anterior and 16 of both leaflets. (40+4+16= 60). 

Answer: thank you very much for this valuable comment: we went through the statistical records. There was an error in our analysis and we do apologize for it; namely 57 patients get chordal replacement, 6 to the anterior, 44 to the posterior and 7 to the anterior and posterior leaflet. This was corrected now in main text L158 to 160   

Comment 3 Table 1: n° 42 (43%) female + 38 (53%) male à total of 80

Answer: thank you very much for your valuable remark. The male proportion was 57% and were 56 patients. This was no corrected in Table 1

Discussion

 Comment 1: It seems that the discussion section is more an explanation of yours surgical strategy rather than an analysis of your results. 

Answer: dear reviewer thank you very much for your valuable remark. indeed, since our approach in Barlow disease is kind of unique tried to explain how we were able to achieve the surgical excellence with implementation of existing literature. We adapted the discussion section accordingly and added new references of most recent reports.

Limitations

Comment 1 Line 325-332: these are not limitations of the study, but they are instead the strong points; so they should be removed from this section. I would had some limitations: 

Patients are hyper selected (mean age 59, mean Euroscore 0,87%, low comorbidity)

Follow-up of last patients don’t reach the 7 years. 

Answer: this section was now positioned as last paragraph in section discussion. The limitation was adapted now where we followed your suggestion of mentioning young population with low risk score as major disadvantages.

Reviewer 2 Report

Comments and Suggestions for Authors

Interesting article

To review the title and some emphatic sentences (sometimes a little self-congratulatory)

Excellent case study

Can we insist more on the comparison with other case studies that make your case study worthy?

The introduction on repair techniques is very interesting. I would add some educational images

Is the reoperation rate for bleeding in range or slightly higher?

I would focus more on the technical choice of not performing the resection of the prolapsing valve leaflet

In how many patients was the Alfieri technique applied?

Author Response

Reviewer 2

Comment 1 Interesting article

To review the title and some emphatic sentences (sometimes a little self-congratulatory)

Excellent case study. Can we insist more on the comparison with other case studies that make your case study worthy?

Answer: thank you very much for your valuable comment. We added a paragraph in discussion where our result are compared to the recent literature. Here with focus on outcome such a success of repair as well on incidence of adverse cardiovascular events. In this setting we added two new references as well.

Comment 2 The introduction on repair techniques is very interesting. I would add some educational images

Answer: thank you very much for this valuable opinion. Unfortunately, we do not have technical images. However, we are very thankful for this input since we plan to publish a manuscript on pure surgical technique.

Comment 3: Is the reoperation rate for bleeding in range or slightly higher?

Answer: thank you very much for your valuable comment. Indeed reoperation rate is slightly higher, this may be in line with very conservative bleed and anticoagulation management in our departement. We now highlighted this part in section discussion last paragraph.

Comment 4I would focus more on the technical choice of not performing the resection of the prolapsing valve leaflet

Answer: thank you very much for this input. We now rephrased this part in section discussion as well.

 Comment 5 In how many patients was the Alfieri technique applied?

Answer: the Alfieri technique was not applied in any of our patients this was as well mentioned in section methodology.

Reviewer 3 Report

Comments and Suggestions for Authors

Dear authors, thank you for submitting this important work.

It deals with an important health issue and the manuscript is generally well-written.

Ethical disclosures are disclosed in a satisfactory fashion.

Some specific comments, as outlined below:

- there is complete lack of echocardiographic characterization of these patients. This data should be put forward. We do not know any important characteristics of these patients like below

- What were average end-systolic volumes and dimensions of LV? 

- How many patients had reduced ejection fraction?

- What were the ejection fractions of these patients?

- What were EROA values of these patients and what were regurgitant volumes in this patient population?

- By the currently available data that authors present, this seems like a low-risk, young population with not too many comorbidities, hence such a high percentage of procedural success. This is important to highlight.

- in the Supplemental Materials, Table 1. authors says that continuous variables are shown as N (%) for categorical variables and median (interquartile range) for continuous variables, however, continuous variables are not shown in the typical interquartile range (it should be shown like 25th and 75th percentile). Please amend this.

- We also do not know anything about the guideline-directed medical therapy in these patients. Nothing of this is disclosed and can affect outcomes on the long run.

- I would refrain from putting statements such as “hundred percent success” or similar in the title of this manuscript because this might be misleading outside of the context of the study. Please rephrase your manuscript differently.

- It should be stated numerically how many different operators performed procedure and were there any periprocedural differences between the operators. Also, it would be beneficial to state were these high volume operators as these results might not be applicable to many other centers worldwide.

Author Response

Dear authors, thank you for submitting this important work.

It deals with an important health issue and the manuscript is generally well-written.

Ethical disclosures are disclosed in a satisfactory fashion.

Some specific comments, as outlined below:

Comment 1 there is complete lack of echocardiographic characterization of these patients. This data should be put forward. We do not know any important characteristics of these patients like below
What were average end-systolic volumes and dimensions of LV? 

How many patients had reduced ejection fraction?

- What were the ejection fractions of these patients?

- What were EROA values of these patients and what were regurgitant volumes in this patient population?

Answer: Thank you very much for your important input. The mean LVEF was 60% with and the most inferior value of the LVEF in our cohort was 51%. All patienst included have had normal LVEF.  The LV dimensions as well the ESV are not presented, namely we are analyzing in overall mitral valve cohort the remodeling of the LV after MV repair. Since the data are not completed yet we are not able to share them in this manuscript

Comment 2 By the currently available data that authors present, this seems like a low-risk, young population with not too many comorbidities, hence such a high percentage of procedural success. This is important to highlight.

Answer: thank your very much for this important input. We addressed this part on P12 to 13 L 283 to 292

Comment 2 in the Supplemental Materials, Table 1. authors says that continuous variables are shown as N (%) for categorical variables and median (interquartile range) for continuous variables, however, continuous variables are not shown in the typical interquartile range (it should be shown like 25th and 75th percentile). Please amend this.

Answer: thank you very much for this remark. The categorical variables are presented as mean and standard deviations, this was now adapted in Table 1

Comment 3 We also do not know anything about the guideline-directed medical therapy in these patients. Nothing of this is disclosed and can affect outcomes on the long run.

Answer: Thank you very much for your valuable remark, adding those result would have additional information. Unfortunately we do not have those information’s.

Comment 4 I would refrain from putting statements such as “hundred percent success” or similar in the title of this manuscript because this might be misleading outside of the context of the study. Please rephrase your manuscript differently.

Answer: thank you very much for this remark. we changed the title now as follows: Excellent result in minimally invasive non resectional mitral valve repair for Barlow mitral valve disease long-term results

Comment 5 It should be stated numerically how many different operators performed procedure and were there any periprocedural differences between the operators. Also, it would be beneficial to state were these high-volume operators as these results might not be applicable to many other centers worldwide.

Answer: thank you very much for this remark. Surgeries were performed by different staff members, since our department is university hospital, and we have a high fluctuation frequency of the staff. However, the senior author DB performed the most interventions. Indeed the procedures are standardized and all staff members are following the same procedures as they are trained in our institution.

Round 2

Reviewer 1 Report

Comments and Suggestions for Authors

Authors fixed the article following the indications

Author Response

Comment 1: Authors fixed the article following the indications

Answer: thank you very much for your valuable comment

Reviewer 3 Report

Comments and Suggestions for Authors

Thank you for revising your manuscript.
All highlighted lack of data etc. such as the lack of echocardiographic indices, pharmacotherapy at baseline etc. must be clearly stated as manuscript limitation within the manuscript. Other comments to which authors did not have suitable response should also be emphasized as limitation in the Discussion section of the manuscript.

Similarly, I would propose that the title of your manuscript is. Long-term results in minimally invasive non-resectional mitral valve repair for Barlow mitral valve disease. Again, authors need to refrain from mundane terms such as "excellent, 100%" etc. This is not appropriate for medical journals.

Author Response

Comment 1: All highlighted lack of data etc. such as the lack of echocardiographic indices, pharmacotherapy at baseline etc. must be clearly stated as manuscript limitation within the manuscript. Other comments to which authors did not have suitable response should also be emphasized as limitation in the Discussion section of the manuscript.

Answer: thank your very much for your valuable opinion. We added the recommended informations now in limitations section. Here focus on lack of information on left ventricular remodelling as well no information on medical treatment in long term.

2 Comment: Similarly, I would propose that the title of your manuscript is. Long-term results in minimally invasive non-resectional mitral valve repair for Barlow mitral valve disease. Again, authors need to refrain from mundane terms such as "excellent, 100%" etc. This is not appropriate for medical journals.

Answer: Title was now changed as proposed. Furtert the words excellent as well 100% were removed and replaced with corresponding statements.